# Hypoxia-Mediated Long Non-Coding RNA Fragment Identified in Canine Oral Melanoma through Transcriptome Analysis

**DOI:** 10.3390/vetsci11080361

**Published:** 2024-08-09

**Authors:** Yasunori Hino, Mohammad Arif, Md Mahfuzur Rahman, Al Asmaul Husna, MD Nazmul Hasan, Naoki Miura

**Affiliations:** 1United Graduate School of Veterinary Science, Yamaguchi University, 1677-1 Yoshida, Yamaguchi 753-0841, Japan; 2Joint Graduate School of Veterinary Medicine, Kagoshima University, 1-21-24 Korimoto, Kagoshima 890-0065, Japan; mdarif38515@bau.edu.bd (M.A.);; 3Joint Faculty of Veterinary Medicine, Kagoshima University, 1-21-24 Korimoto, Kagoshima 890-0065, Japan,

**Keywords:** hypoxia, melanoma, ncRNA, NGS, dog, metastasis

## Abstract

**Simple Summary:**

Hypoxia drives the dysregulation of RNA molecules in canine oral melanoma (COM), influencing tumor progression and metastasis. While hypoxia-associated miRNAs have been explored, this study focused on other non-coding RNAs (ncRNAs) using NGS and qPCR. The present study revealed significant hypoxia-regulated alterations in ncRNA expression profiles, highlighting ENSCAFT00000084705.1 as consistently downregulated in COM tissues and cell lines compared to healthy tissue. Notably, its absence in plasma and extracellular vesicles suggests limited biomarker potential. This study provides evidence of transcriptional changes in ncRNAs except for miRNAs in COM, highlighting ENSCAFT00000084705.1 as a promising candidate for future investigations into the role of the transcriptome in the hypoxia-driven progression of this aggressive cancer.

**Abstract:**

Hypoxia contributes to tumor progression and metastasis, and hypoxically dysregulated RNA molecules may, thus, be implicated in poor outcomes. Canine oral melanoma (COM) has a particularly poor prognosis, and some hypoxia-mediated miRNAs are known to exist in this cancer; however, equivalent data on other hypoxically dysregulated non-coding RNAs (ncRNAs) are lacking. Accordingly, we aimed to elucidate non-miRNA ncRNAs that may be mediated by hypoxia, targeting primary-site and metastatic COM cell lines and clinical COM tissue samples in next-generation sequencing (NGS), with subsequent qPCR validation and quantification in COM primary and metastatic cells and plasma and extracellular vesicles (EVs) for any identified ncRNA of interest. The findings suggest that a number of non-miRNA ncRNA species are hypoxically up- or downregulated in COM. We identified one ncRNA, the long ncRNA fragment ENSCAFT00000084705.1, as a molecule of interest due to its consistent downregulation in COM tissues, hypoxically and normoxically cultured primary and metastatic cell lines, when compared to the oral tissues from healthy dogs. However, this molecule was undetectable in plasma and plasma EVs, suggesting that its expression may be tumor tissue-specific, and it has little potential as a biomarker. Here, we provide evidence of hypoxic transcriptional dysregulation for ncRNAs other than miRNA in COM for the first time and suggest that ncRNA ENSCAFT00000084705.1 is a molecule of interest for future research on the role of the transcriptome in the hypoxia-mediated progression of this aggressive cancer.

## 1. Introduction

Canine oral melanoma (COM) research commands a high priority in veterinary medicine. Melanomas represent around seven percent of all malignant tumors in dogs, and most frequently occur in the oral cavity [1,2,3]. COM has the poorest prognosis of any melanoma in dogs with a median survival time for stage II and stage III of the disease at around six to eight months from diagnosis [3]. Moreover, the survival time is reduced to two months if pulmonary metastasis develops at the time of the diagnosis of COM [3]. COM research may have a wider application since these canine tumors resemble some human mucosal melanomas macroscopically [4] and human non-UV-induced cutaneous melanomas clinically and histopathologically [5]. Advances in the diagnosis and treatment of this disease in dogs may ultimately benefit human patients [2,6].

COM is regarded as one of the most aggressive tumors in dogs, and a hypoxic tumor microenvironment may be implicated in its highly aggressive nature and high propensity to metastasize [7]. As a solid tumor, COM sees a fall in oxygen pressure (from levels of 1% to 2%), with the oxygen supply proving unequal to the tumor growth [8,9]. Crucially, hypoxia can cause the upregulation of tumorigenic factors and downregulation of anti-tumorigenic factors, in addition to mediating tumor angiogenesis and cancer stem cell quiescence [10,11,12]. Thus, in a number of ways, hypoxia may act as a driver of tumor aggression and promote the progression to metastasis and is particularly relevant for highly aggressive and frequently metastasizing malignant tumors such as COM.

One promising line of COM research involves identifying elements of the transcriptome that may be dysregulated hypoxically. Changes in long non-coding RNA (lncRNA) profiles have been reported [13] in metastatic COM. Hypoxia is known to alter gene expression and RNA profiles within the tumor microenvironment [14], and dysregulated molecules may thus have the potential for utility as therapeutic targets or diagnostic biomarkers. We previously identified a number of molecules of interest based on dysregulated expression, which encompasses snoRNA, snRNA, piRNA, and tRNA fragments (tRFs) [15] in tumor tissue from COM patients. In further reports, we have highlighted dysregulated miRNAs [16], lncRNA, and tRFs [17] as exosomal biomarkers in plasma from COM patients. However, to the best of our knowledge, previous molecular investigations in COM have involved hypoxically dysregulated miRNAs [9] and Y-RNA [7]. A comprehensive understanding of the hypoxic mechanisms underlying COM progression and metastasis requires evidence from the entire RNA transcriptome. Thus, further investigations of non-coding RNAs (ncRNAs) other than miRNAs are needed. Recent advances in molecular analysis technologies have added to the identification of transcriptome elements, with next-generation sequencing (NGS) playing a particularly important role in disease diagnosis and progression.

Against this background, we set out in this study to investigate the hypoxic mediation of non-miRNA ncRNAs in COM by identifying hypoxically deregulated ncRNAs through the comparative NGS analysis of primary-site and metastatic COM cell lines, COM tissue samples, and oral tissue from healthy dogs. Any ncRNA identified as a target of interest was then validated for its expression level using qPCR assays. Furthermore, we evaluated the target of interest as a potential biomarker by measuring its levels in plasma and plasma extracellular vesicles (EVs).

## 2. Materials and Methods

### 2.1. Clinical Samples

Tumor tissue and blood samples were collected from dogs with COM undergoing treatment in the Kagoshima University Veterinary Teaching Hospital (KUVTH) or an affiliated veterinary clinic between 2014 and 2022. Oral melanoma tissue samples were collected from thirty dogs (*n* = 30). Among the 30 COM patients, twenty plasma samples (*n* = 20) were obtained. In addition, five control blood samples (*n* = 5) were collected from COM-free dogs. Definitive diagnoses were made after histopathological examination by certified pathologists employed by KUVTH. All procedures involving animals received approval from the animal ethics committee at KUVTH (Approval No. KVH220001). The study population and sample information are presented in Table 1. Briefly, the sample donors included 20 males and 10 females, with a median age of 12 years (range: 7–16 years), and were drawn mainly from small breeds. The healthy controls were beagles that had been purpose-bred for research at Shin Nippon Biomedical Laboratories, Ltd., Drug Safety Laboratories (Kagoshima, Japan).

Consent for sample collection was obtained from the owner of each dog. The tissue samples were transferred to RNAlater immediately after collection for storage. Blood samples were centrifuged to obtain plasma in accordance with our standard protocol. Blood samples were collected in tubes containing 3.2% of the sodium citrate anticoagulant and then centrifuged for 10 min at 3000× *g* immediately after collection to separate plasma from other cellular elements such as red blood cells (RBCs). The supernatant was then carefully aspirated and subjected to additional high-speed centrifugation at 16,000× *g* for 10 min at 4 °C to remove residual cellular debris or platelets. Pure plasma was collected from the supernatant without disturbing the pellet. All samples were stored at −80 °C until analysis.

### 2.2. Cell Lines and Cell Culture

KMeC and LMeC are canine oral melanoma cell lines originating from primary and metastatic tumor sites, respectively, and were preserved in a freezing medium (039-23511, CultureSure, Fujifilm Wako Pure Chemical Corporation, Osaka, Japan). The cell lines were cultured under previously described conditions [18]. In brief, cells were cultured in a medium containing Roswell Park Memorial Institute (RPMI) media-1640 (Gibco), 10% fetal bovine serum (BI, Biological Industries, Beit Haemek, Israel), 1% L-glutamine solution (Fujifilm Wako Pure Chemical Corporation, Osaka, Japan) and antibiotics (penicillin-streptomycin, Sigma; 100 units/mL of penicillin and 100 μg/mL of streptomycin). Cells were cultured (biological replicates/line: 6) in parallel under normoxic (5% CO_2_, 21% O_2_, 37 °C) or hypoxic (5% CO_2_, 2% O_2_, 37 °C) conditions for the specified duration. A hypoxic environment was achieved using a hypoxic incubator (HeracellTM 150i, Thermo Fisher Scientific, Waltham, MA, USA). The culture medium was changed at 48 h intervals. Cells were grown until confluency. Cells were counted using an automated cell counter (LUNA-II^TM^ automated cell counter, Annandale, Virginia, USA). Cell lysates were obtained at relevant time intervals, and RNA was subsequently extracted from each lysate for ncRNA expression analysis. Cell experiments were conducted twice to ensure reproducibility. To explore the impact of hypoxia on melanoma, cells were cultured for up to 96 h. We set the cell lysate collection times at 12, 48, and 96 h so as to mitigate any immediate effect of medium change.

### 2.3. Isolation of EVs

EVs were isolated using the Total Exosome RNA and Protein Isolation Kit (Invitrogen, Waltham, MA, USA, Thermo Fisher Scientific) from plasma in accordance with the manufacturer’s instructions and the procedure described earlier [17]. In summary, a 300 µL aliquot from the plasma sample was thoroughly mixed with a half-volume 1 × PBS, followed by the addition of 90 µL of the exosome precipitation reagent. The resultant mixture was thoroughly vortexed and then centrifuged at 10,000× *g* for 5 min. The supernatant was discarded to isolate the pellet containing the EVs. Subsequently, 150 µL of 1 × PBS was added to resuspend the pellet, which was then stored at −80 °C until analysis.

### 2.4. RNA Isolation from Cells, Plasma, and Plasma EVs

Total RNA was extracted from the COM cells and tissues using a mirVana™ miRNA Isolation Kit (Thermo Fisher Scientific, Vilnius, Lithuania) and from plasma and plasma EVs using a mirVana PARIS Kit (Thermo Fisher Scientific, Vilnius, Lithuania), following the instructions provided by the manufacturer. Prior to RNA extraction, 5 μL of synthetic cel-miR-39 was mixed with each plasma or plasma-derived EV sample in order to normalize the differences in expression. Briefly, each cell lysate or plasma sample was mixed with an equal amount of the 2× denaturation solution. The miRNA homogenate was added at a ratio of 10:1, and the mixture was then incubated at 4 °C for 10 min, after which acid–phenol/chloroform (Ambion LTD, Huntingdon, Cambridgeshire, UK) was added at a volume equal to that of the initial cell lysate. The resultant mixture was vortexed thoroughly and then centrifuged at 15,000× *g* for 5 min at 25 °C. After centrifugation, the supernatant was carefully removed and mixed with 1.25 volumes of ethanol (99.9% purity) in an Eppendorf tube. The tube was then centrifuged using the spin column provided with the kit to trap the RNA particles on the filter paper. Finally, RNA was eluted from the filter membrane by the elution solution pre-heated to 95 °C. The concentration of total RNA was measured using the NanoDrop 2000c spectrophotometer (Thermo Fisher Scientific, Wilmington, DE, USA). RNA quality and integrity were assessed using an Agilent 2100 Bioanalyzer (G2939BA, Agilent Technologies, Santa Clara, CA, USA). The RNA Integrity Numbers (RINs) for the cell samples ranged from 8.5 to 9.5.

### 2.5. Next-Generation Sequencing

NGS targeting total RNA was commissioned by a specialist genetics analysis laboratory (Hokkaido System Science, Hokkaido, Japan) and conducted as described previously [9]. Briefly, small RNA libraries were prepared using the TruSeq Small RNA Library preparation kit from 1 µg of total RNA, following the manufacturer’s instructions (Illumina, San Diego, CA, USA). Adapters (5′ and 3′) were then ligated to the small RNAs, and cDNA was synthesized using reverse transcription, followed by substantial amplification. The amplified cDNA was subjected to gel electrophoresis to check its purity and then sequenced using an Illumina/Hiseq2500 system by Hokkaido System Science (Hokkaido, Japan). The sequencing yielded high-quality reads with Phred scores > 35. Finally, the sequences were stored in the NCBI Sequence Read Archive (SRA) database with the accession number PRJNA629070.

NGS data were obtained from the analyses of canine oral healthy tissue (control tissues, *n* = 3), clinical COM tissue (*n* = 8), and KMeC and LMeC cell lines (*n* = 3 biological replicates for each cell line in either normoxic or hypoxic conditions, cultured for 48 h), under normoxic and hypoxic conditions. The sequence reads were stored in the Sequence Read Archive (SRA) (www.ncbi.nlm.nih.gov/sra, accessed on 11 March 2023) with the accession number PRJNA629070. A summary of the NGS data analysis has already been documented in previous studies [9]. NGS data were trimmed, quality checked, and underwent subsequent analysis, as described in earlier studies [7,9]. miRBase and Ensembl databases were utilized to annotate the reads. All cell line replicates were clustered together under normoxic and hypoxic conditions, suggesting they are appropriate for further differential expression analysis [9].

### 2.6. Bioinformatic Analysis

NGS reads were examined using the CLC Genomics Workbench, V10.0 and 12.0, in accordance with the developer’s instructions (https://digitalinsights.qiagen.com, CLC Bio, Qiagen, Germany, accessed on 15 March 2023). The final analysis preceded adapter trimming, QC checking, and the sorting of ambiguous reads. Parameters were set as the recommended defaults for all analysis runs. Initially, adapters and low-quality, ambiguous reads were removed. Small RNA sequences were extracted and counted from the clean reads. Subsequently, the annotation of the extracted reads was obtained using miRBase and canine and human non-coding RNA databases from ENSEMBL (Canis_familiris/canfam3.1.ncrna and Homo_sapiens/GRCh37.ncrna). A previous study from our laboratory reported miRNAs annotated from miRbase [9]. Non-miRNA ncRNAs were investigated in this study. Sequence counts were regarded as the expression values for ncRNAs. The empirical analysis of differential gene expression (EDGE) was used to identify any significantly dysregulated ncRNAs based on the following criteria: FDR *p*-value < 0.05, |FC| > 2, and minimum expression > 10 [mature count] per replicate.

### 2.7. qPCR

The expression level of the target ncRNA was measured using TaqMan gene expression assays (Thermo Fisher Scientific, Waltham, MA, USA). qPCR was performed following previously described procedures [7]. First, cDNA was prepared from 2 ng of the RNA sample using a TaqMan MicroRNA Reverse Transcription kit following the manufacturer’s protocol (4366597, Thermo Fisher Scientific, Vilnius, Lithuania). In the next step, qPCR was performed with a StepOnePlus real-time PCR system (Thermo Fisher Scientific, Woodlands, Singapore) using a TaqMan Fast Advanced Master Mix kit (4444557, Thermo Fisher Scientific, Vilnius, Lithuania). The thermocycling conditions for qPCR were as follows: 50 °C for 2 min, 95 °C for 20 s, followed by 40 cycles of 1 s denaturation at 95 °C and 20 s annealing/extension at 60 °C. RNU6B, miR-16, and miR-186 were used as internal controls for the evaluation of the relative expression of the targeted ncRNA molecule in cell lines, plasma, and plasma EVs, respectively [16]. The final expression level was calculated following the 2^−ΔΔCT^ method [19]. A qPCR test providing a cycle threshold (Ct) value greater than 35 was regarded as undetected for that specified sample. All qPCR assays were performed twice to confirm their reproducibility. Expression values of the control miRNAs (miR-16 for plasma and miR-186 for plasma EVs) were evaluated using TaqMan microRNA assays. The primer IDs for the internal controls were RNU6B: 001093, miR-16: 000391, and miR-186: 002285. Information on each primer can be found at https://www.thermofisher.com/order/genome-database/ (accessed on 8 July 2023). Primers for the selected ncRNA (Ensembl ID; ENSCAFT00000084705.1; sequence: 5′- ATTCCTGGACTCACGGATACT-3′) was custom-designed.

### 2.8. Statistical Analysis

Differences between groups were statistically assessed using the Kruskal–Wallis test and Mann–Whitney U test to evaluate the relative expression of the target ncRNA. A statistical test with a *p*-value of less than 0.05 (*p* < 0.05) was considered significant. All statistical analyses and graph visualizations were conducted using GraphPad Prism 9 (https://www.graphpad.com/, accessed on 10 September 2023).

## 3. Results

### 3.1. NGS Profiling of Hypoxia-Mediated ncRNAs

To initially determine ncRNAs that are differentially expressed in clinical COM tissue, and primary-site (KMeC) and metastatic (LMeC) COM cells versus healthy oral tissue, and in hypoxic versus normoxic KMeC and LMeC cells, we applied stringent filtering criteria to NGS reads (FDR *p*-value < 0.05, |FC| > 2, and minimum expression > 10 [mature count] per replicate), and listed the ncRNAs thus classified as upregulated or downregulated. The numbers of differentially expressed ncRNAs for all comparisons are shown in Figure 1.

Focusing on hypoxia versus normoxia comparisons in COM cells, we identified 928 and 1367 hypoxically dysregulated ncRNAs for KMeC and LMeC cells, respectively. For KMeC cells, 242 of these ncRNAs were upregulated, and 686 were downregulated, and for LMeC cells, 266 were upregulated, and 1101 were downregulated (Figure 1, Appendix A). These findings suggest that hypoxia mediates ncRNA expression in COM.

### 3.2. Identification of Target ncRNAs

To identify ncRNAs of interest, we then subjected data on differentially expressed ncRNAs to Venn diagramming to find any molecules showing a consistent pattern across clinical COM tissue and COM cell lines. Comparing the expression between hypoxia and normoxia in cell lines may not lead to the sufficiently robust identification of target molecules because of the possibility for developing unintended hypoxia in the cultured cells. Accordingly, we set out to identify hypoxically dysregulated, melanoma-specific target ncRNAs, which were commonly up- or down-regulated in clinical COM tissue, hypoxic KmeC, and LMeC cells versus healthy oral tissue, and hypoxic KMeC and LMeC cells versus normoxically cultured equivalents. Target ncRNAs were identified as those in the intersection of three comparisons in the relevant Venn diagram. The results are visually presented, separating upregulated and downregulated ncRNAs in Figure 2, with a full breakdown of numbers in Appendix A.

Although 16 ncRNAs were commonly upregulated in clinical COM tissue and hypoxic KMeC or LMeC versus healthy oral tissue or the relevant normoxic cells, none of them satisfied the criterion for selection as a target ncRNA because they were not upregulated in LMeC (metastatic) vs. KMeC (primary site) COM cells.

By contrast, one ncRNA (ENSCAFT00000084705.1) was commonly downregulated in COM tumor tissue and hypoxic KMeC or LMeC versus healthy oral tissue or the relevant normoxic cells in NGS (Figure 2) and, thus, satisfied the criteria for selection as a target ncRNA based on downregulation in LMeC vs. KMeC cells. The relative expression levels of the target ncRNA (ENSCAFT00000084705.1) across all types of NGS samples are depicted in Figure 3.

### 3.3. qRT-PCR Validation of Downregulated ncRNA (ENSCAFT00000084705.1)

#### 3.3.1. Relative Expression in Hypoxic KMeC and LMeC Cells

To validate our NGS findings, we investigated the expression of ENSCAFT00000084705.1 at three different time intervals (12, 48, and 96 h) in hypoxic KMeC and LMeC cells. The expression level decreased significantly (*p* < 0.05) in both cells at 48 and 96 h under hypoxic conditions compared to 12 h (Figure 4A,B). Moreover, expression significantly differed (*p* < 0.01) between LMeC and KMeC cells at 96 h (Figure 4C), showing a non-significant trend that was similar at 48 h (*p* > 0.05). Accordingly, ENSCAFT00000084705.1 expression appeared to decrease progressively with hypoxia, and the magnitude of this decrease was greater in metastatic (LMeC) cells compared to primary-site (KMeC) cells.

#### 3.3.2. Relative Expression in Plasma and Plasma-Derived EVs

We further examined the expression of ncRNA in plasma and plasma-derived EVs from clinical COM patients. Although the internal control (miR-186) was detectable in plasma EVs, ENSCAFT00000084705.1 was undetectable in both plasma and plasma EVs (Figure 5). The cycle threshold (Ct) values during qPCR validation surpassed 35, failing to meet the criteria for our experimental conditions. Thus, our results imply that ENSCAFT00000084705.1 was either absent or minimally expressed in plasma and plasma EVs.

## 4. Discussion

In this study, we aimed to build on previous research on RNA molecules, evaluate the expression of ncRNAs other than miRNAs in COM, and investigate their links to hypoxia, which may be heavily implicated in the progression and metastasis of this cancer. To the best of our knowledge, this is the first report on such hypoxic dysregulation of non-miRNA ncRNAs.

In the initial findings of our study, we demonstrated transcriptional dysregulation for non-miRNA ncRNAs, with 928 (242 up and 686 downregulated) and 1367 (266 up and 1101 downregulated) relevant molecules in hypoxically cultured KMeC and LMeC cells, versus their normoxic counterparts, respectively, based on NGS results. Lou et al. (2024) similarly revealed the dysregulation of ncRNAs due to hypoxia, identifying 2455 hypoxia-induced lncRNAs from the human breast cancer dataset [20]. Thus, the expression of ncRNAs appears to be linked with hypoxic dysregulation. Moreover, the number of dysregulated ncRNAs was greater in metastatic than primary-site COM cells, suggesting these species might exhibit greater efficacy in coping with hypoxic cellular stress in the metastatic stage.

As this study is the first to investigate hypoxia-regulated ncRNAs in COM, the findings cannot be compared to those from previously conducted studies involving patients of different ages, breeds, or geographical locations. However, the findings for non-miRNA ncRNAs from the present study broadly reflect the extent of dysregulation we previously identified for miRNAs in comparable screening experiments [9]. The functional relationships between miRNAs and other ncRNAs have yet to be elucidated; however, considering that hypoxia is known to alter gene expression in multiple ways [21,22,23], our findings may indicate that RNA molecules across the whole transcriptome are affected by or may interact in some way with a hypoxic tumor microenvironment. We consider that non-miRNA ncRNAs are thus also of potential interest to veterinary oncologists seeking to better understand why COM is such an aggressive cancer.

From among the hypoxically dysregulated ncRNAs, we then set out to find molecules of interest that showed a broadly consistent pattern, either upregulation or downregulation, across the COM cell lines and tumor tissue samples in comparison with healthy oral tissue. COM tumor tissue samples were included in these comparisons to safeguard against misidentifying a transcriptome element as dysregulated due to any unintended hypoxia that might occur during cell culture. As a very aggressive, solid cancer, COM develops hypoxic microenvironments quickly, which plays a role in metastasis [24,25]. Thus, the final part of the consistent pattern we were seeking involved the trend of upregulation or downregulation being evident in the metastatic cell line versus the primary-site cell line (LMeC vs. KMeC cells). Eventually, we were only able to identify one candidate ncRNA (ENSCAFT00000084705.1), which was commonly downregulated in every relevant comparison (no candidate satisfied the criteria for the upregulation pattern). These findings differ somewhat from our previously reported results with miRNAs, where we found 15 (14 upregulated and 1 downregulated) commonly and hypoxically dysregulated miRNAs in COM [9] evaluated with the same criteria used in this study. Taken together, these results may tentatively indicate that miRNA species show sensitivity to a hypoxic microenvironment in greater numbers than other ncRNA species, but considerable further research is needed.

Although ENSCAFT00000084705.1 was the sole target ncRNA we identified, its association with hypoxia appears robust. This ncRNA was downregulated in COM cells and tumor tissue versus healthy oral tissue and in hypoxic cells versus normoxic cells. Furthermore, its downregulation in metastatic cells was consistent with a progressive trend of downregulation as COM, an aggressive cancer with a high propensity to metastasize, progresses. We were able to validate our NGS findings for this ncRNA in PCR assays with KMeC and LMeC cells and demonstrated a chronologically progressive downregulation of greater magnitude in metastatic cells. We thus speculate that ENSCAFT00000084705.1 plays a distinct and important role in the hypoxic COM microenvironment that is linked to tumor progression and metastasis [26,27]. Hypoxia is known to cause the downregulation of anti-tumorigenic factors like p53, dicer, e-cadherin, and so on [28,29,30], indicating that ENSCAFT00000084705.1 may have a role in the development of future therapeutic strategies.

Having established ENSCAFT00000084705.1 as an ncRNA of interest, we then set out to evaluate its level in plasma and plasma EVs. As EVs may carry cargo from tumor cells to other regions in the body [31], their target molecule concentrations may reveal any potential role in the spread of a tumor beyond its primary site [32]. However, ENSCAFT00000084705.1 proved almost undetectable in plasma and plasma EVs in qPCR assays, and we deduced that it is either absent or minimally expressed in these matrices. Its expression could, thus, appear to be tissue-specific, and we regard ENSCAFT00000084705.1 as having little promise as a biomarker for COM generally or for metastatic versus non-metastatic COM. This contrasts with our findings in a previous study on another lncRNA (ENSCAFT00000069599.1), which we suggested is a potential exosomal biomarker for differentiating COM cases from healthy dogs [17].

Our findings add to the understanding of the transcriptional factors in COM. Transcriptome analysis is typically further advanced in humans compared to canine medicine. It is now well established that non-coding RNAs are implicated in the initiation and progression of a number of human cancers, with hypoxic tumor microenvironments suggested to play a role in many cases, with the modulation of the hypoxia/HIF pathway being one part of a possible mechanism [33]. Aberrant lncRNA expression has also been found in human melanoma [34], for which it has been implicated in tumor formation and progression [35]. Even though thousands of lncRNAs have been identified in dogs [13], their role in canine cancers remains poorly understood, and few have been pinpointed for roles in COM. Our identification of ENSCAFT00000084705.1 as a hypoxia-mediated ncRNA is, thus, a useful addition to a still relatively small base of evidence.

The current study has a number of limitations. Firstly, the hypoxic levels of the tissue samples used for NGS analysis were not investigated. However, we believed that the tumor mass was large enough to develop a hypoxic condition inside it. Secondly, clinical validation was conducted with a relatively small population, and our findings on target ncRNA thus require further validation with a larger sample size to draw more robust conclusions. Furthermore, our control dogs were not matched by age, sex, or breed with the main study population. Further studies are needed to validate the clinical usefulness with an increased sample size in which control dogs reflect the age range, sex distribution, and range of breeds in the tumor-bearing population. Finally, there is a need to explore the genes targeted by this lncRNA and their associated signaling pathways. Additionally, investigating the underlying molecular mechanisms and functional role of ENSCAFT00000084705.1 in melanoma development and progression is essential.

## 5. Conclusions

The present study investigated hypoxically dysregulated non-miRNA ncRNAs from COM for the first time, and eventually, our findings revealed that hypoxia may contribute to the dysregulation of ncRNA expression in COM. Our results here allowed us to identify one ncRNA, ENSCAFT00000084705.1, as a hypoxia-mediated factor in COM. Although it may lack potential as a diagnostic biomarker, we consider that ENSCAFT00000084705.1 is a molecule of interest for future studies on this aggressive cancer in dogs.

## Figures and Tables

**Figure 1 vetsci-11-00361-f001:**
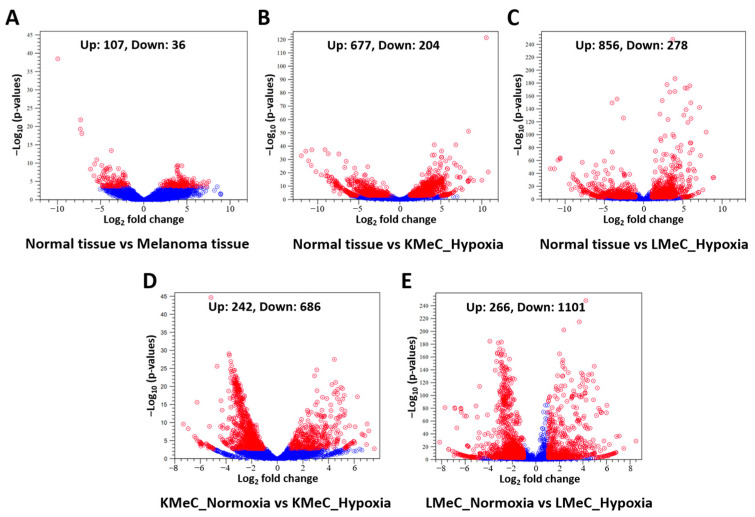
Differentially expressed ncRNAs (except miRNAs) in melanoma tissues and cell lines. (**A–E**) Volcano plot of differentially expressed upregulated and downregulated ncRNAs (except miRNAs) in normal vs. COM tissue (**A**), normal tissue vs. KMeC_hypoxia (**B**), normal tissue vs. LMeC_hypoxia (**C**), KMeC_normoxia vs. KMeC_hypoxia (**D**), and LMeC_normoxia vs. LMeC_hypoxia (**E**). Red dots represent up- and downregulated ncRNAs, and blue dots represent ncRNAs that were not differentially expressed.

**Figure 2 vetsci-11-00361-f002:**
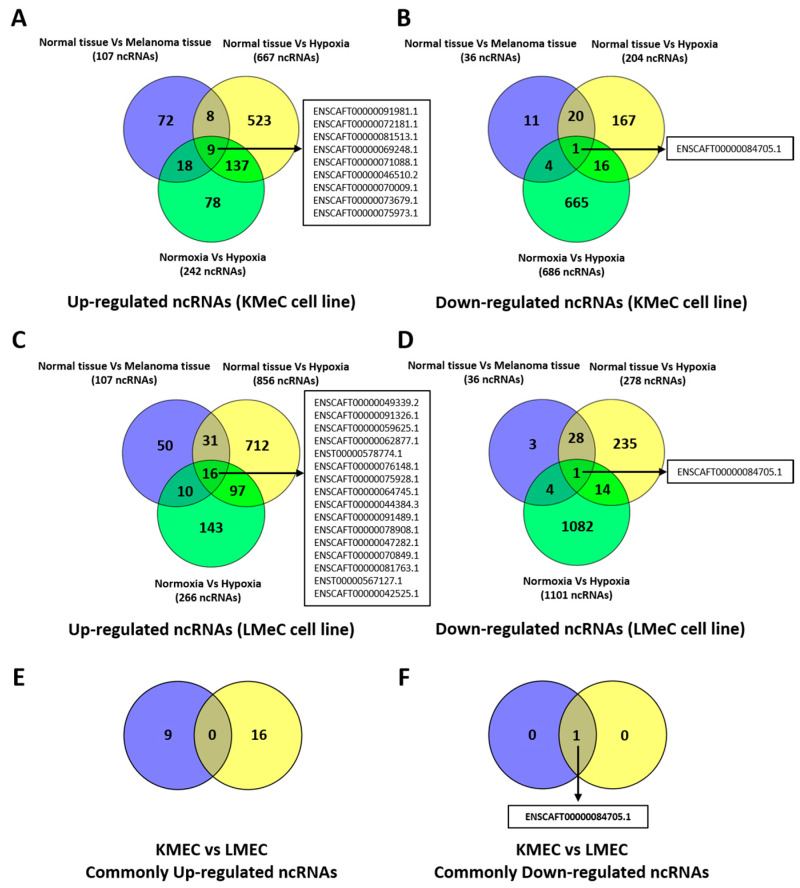
Hypoxia-regulated ncRNAs (except miRNAs) in canine oral melanoma (COM). (**A**,**B**) Venn diagram showing hypoxia-regulated up- and downregulated ncRNAs in KMeC cells. (**C**,**D**) Hypoxia-regulated up- and downregulated ncRNAs in LMeC cells. (**F**) Hypoxia-induced commonly expressed downregulated ncRNAs in both cells. No upregulated ncRNAs were commonly expressed in both cell lines (**E**). Number in the Venn diagram indicate the numbers of dysregulated ncRNAs, whereas the arrows indicate corresponding lists of ncRNAs shared between or among the groups.

**Figure 3 vetsci-11-00361-f003:**
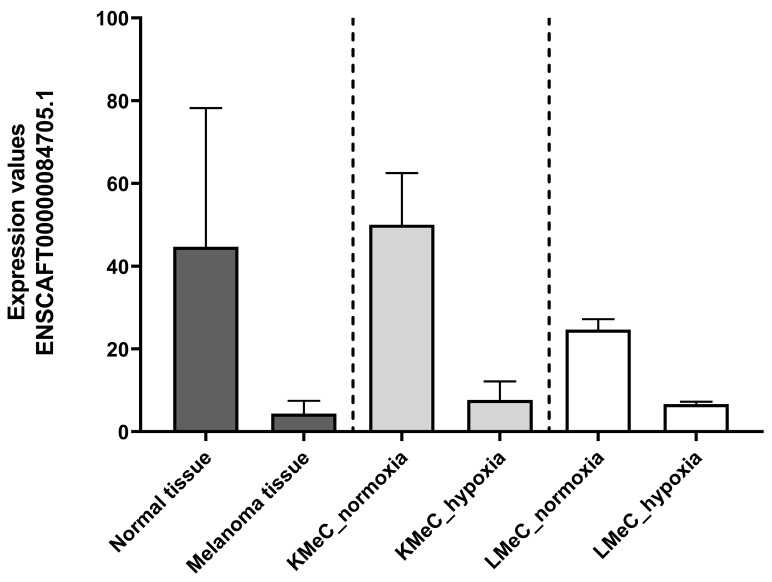
Expression levels of the targeted ncRNA (ENSCAFT00000084705.1) in healthy oral tissue (*n* = 3), melanoma tissue (*n* = 8), normoxic KMeC cells (*n* = 3), hypoxic KMeC cells (*n* = 3), normoxic LMeC cells (*n* = 3), and hypoxic LMeC cells (*n* = 3) in NGS data.

**Figure 4 vetsci-11-00361-f004:**
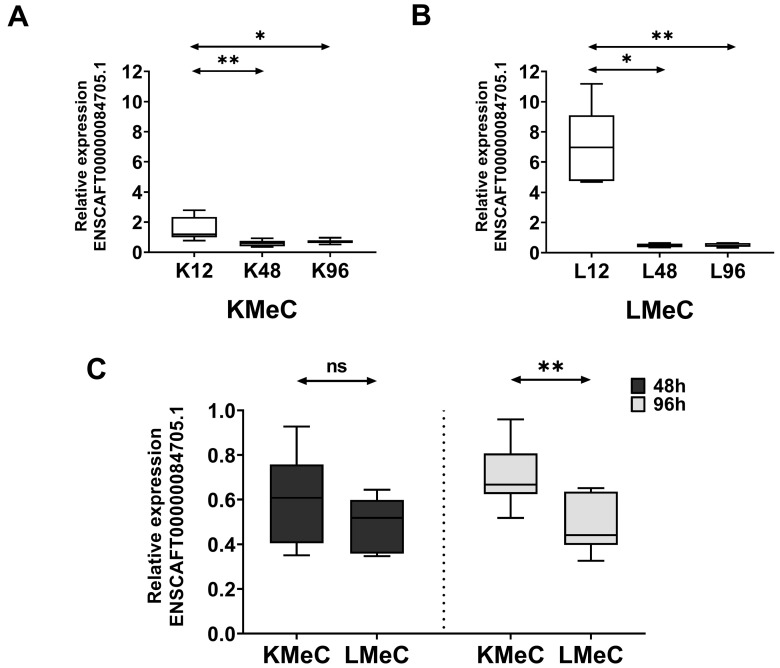
Relative expression of targeted ncRNA (ENSCAFT00000084705.1) at different time intervals in hypoxic KMeC and LMeC cells identified by qRT-PCR. Relative expression of ENSCAFT00000084705.1 at 12, 48, and 96 h in hypoxic KMeC cells (*n* = 6, for each time point) (**A**) and LMeC cells (*n* = 6, for each time point) (**B**). (**C**) Relative expression level of targeted ncRNA (ENSCAFT00000084705.1) between primary (KMeC, *n* = 6) and metastatic (LMeC, *n* = 6) COM cell lines under hypoxic conditions. The dark grey color box indicates the relative expression level of ENSCAFT00000084705.1 in KMeC and LMeC cells after 48 h, and the light grey color represents the expression levels after 96 h. The *y*-axis in the graph indicates the relative expression levels of ENSCAFT00000084705.1. The difference between each time point in both cells was analyzed using the Kruskal–Wallis test, whereas the Mann–Whitney U test was utilized to statistically analyze the difference between KMeC and LMeC cells at different time intervals. A *p*-value of less than 0.05 (*p* < 0.05) was considered statistically significant. * denotes a significant test (* = *p* < 0.05 and ** = *p* < 0.01); ns = non-significant test.

**Figure 5 vetsci-11-00361-f005:**
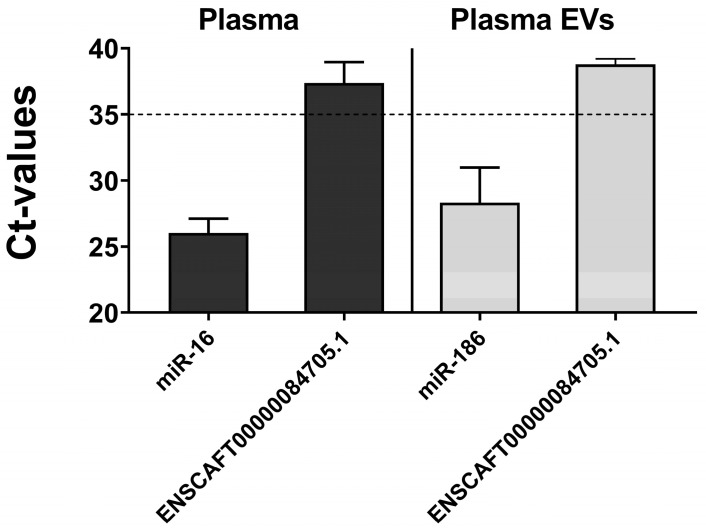
Relative expression of targeted ncRNA (ENSCAFT00000084705.1) in plasma and plasma EVs identified by qRT-PCR. The *y-axis* in the graph indicates the cycle threshold (Ct) values of miR-16 (internal control for plasma, *n* = 5) and ENSCAFT00000084705.1 (*n* = 20) in plasma samples, and miR-186 (internal control for plasma EVs, *n* = 5) and ENSCAFT00000084705.1 (*n* = 11) in plasma EVs. The average Ct value of ENSCAFT00000084705.1 was observed higher than 35 in both plasma and plasma EVs samples.

**Table 1 vetsci-11-00361-t001:** Canine oral melanoma tissue and plasma sample information.

No	Age (Years)	Sex	Breed	Tumor Stage	Metastasis Status	Types of Samples
1	13.3	Male	M.D.	I	—	Tissue, Plasma
2	10.3	Male	Yorkshire	IV	—	Tissue, Plasma
3	10.2	Male	Chiwawa	IV	P	Tissue, Plasma
4	12.7	Male	M.D.	IV	P	Tissue, Plasma
5	14.8	Male	Mongrel	IV	P	Tissue, Plasma
6	10	Male	Golden Retriever	IV	—	Tissue, Plasma
7	10.11	Male	M.D.	I	—	Tissue
8	7.11	Male	M.D.	I	—	Tissue, Plasma
9	10.9	Male	M.D.	IV	—	Tissue, Plasma
10	12	Male	Shiba	IV	P	Tissue
11	13	Male	Pomerania	I	—	Tissue
12	16.3	Male	M.D.	IV	P	Tissue, Plasma
13	11	Male	M.D.	IV	P	Tissue, Plasma
14	12	Male	Mongrel	I	—	Tissue, Plasma
15	11.1	Male	M.D.	IV	P	Tissue, Plasma
16	15.6	Male	Pomeranian	II	P	Tissue, Plasma
17	12.11	Male	M.D.	IV	P	Tissue
18	12.4	Male	Shiba	IV	P	Tissue
19	10.8	Male	M.D.	IV	—	Tissue
20	15.2	Male	Shiba	I	—	Tissue
21	12.4	Female	M.D.	IV	—	Tissue, Plasma
22	14.6	Female	M.D.	II	P	Tissue
23	15.2	Female	Mongrel	IV	—	Tissue
24	15.2	Female	Mongrel	IV	P	Tissue
25	8.2	Female	M.D.	IV	P	Tissue, Plasma
26	15.3	Female	Mongrel	I	—	Tissue, Plasma
27	15.3	Female	Mongrel	I	—	Tissue, Plasma
28	11.8	Female	M.D.	I	—	Tissue, Plasma
29	14	Female	Dalmatian	II	P	Tissue, Plasma
30	12.1	Female	Toy poodle	IV	P	Tissue, Plasma

(M.D.) indicates “Miniature Dachshund”, (P) indicates “Present”, and (—) indicates “Absent”.

## Data Availability

Sequence reads of this study were stored in the SRA (Available online: www.ncbi.nlm.nih.gov/sra (accessed on 11 March 2023). database. (Accession number; PRJNA629070). Other data are included within this article and the Appendix A.

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
