# Peer review of "Hypoxia-Mediated Long Non-Coding RNA Fragment Identified in Canine Oral Melanoma through Transcriptome Analysis"

_vetsci, 2024, doi:10.3390/vetsci11080361_

Round 1

Reviewer 1 Report

Comments and Suggestions for Authors

The authors performed NGS on primary and metastatic COM tissues and cell lines to specifically assess hypoxia-mediated non-miRNA ncRNAs, followed by qPCR validation in COM primary and metastatic cells, plasma and EVs. They identified a number of non-miRNA ncRNA species are hypoxically up- or downregulated in COM, and selected the long ncRNA fragment ENSCAFT00000084705.1, as a ‘molecule of interest’ as it was down-regulated in COM tissues, as well as cultured primary and metastatic cell lines (hypoxic and normoxic), when compared to the oral tissues from healthy dogs.

The manuscript is very well written and the research topic is of interest. However, I have some important concerns.

·      I have concerns with the ‘normal’ tissue samples that were used. They were all from one breed (Beagles) and were from research animals (laboratory animals). Thus they do not represent appropriate controls for the COM tissue samples which came from dogs of many different breeds and who lived with owners (thus had potential environmental exposures not experienced by the laboratory animals). Appropriate controls would have been normal/healthy tissue from the same animal (such as peri-lesional oral tissue). Were the controls dogs of different ages and sexes as the dogs with COM were? No details have been given.

·      Only one primary and one metastatic COM cell line were used. There are others available and it is important to show that the findings are not just specific to these two cell lines.

·      Lines 397-90: simply because ENSCAFT00000084705.1 is expressed in humans does not mean there is a possible “link” between canine and human melanomas. This is too much of an extrapolation and this sentence should be removed unless there is any evidence that ENSCAFT00000084705.1 is expressed in human melanoma.

Some clarification is required in places:

·      In addition, the methods state “n = 3 replicates for each cells in either normoxic or hypoxic condition”. Do these represent technical replicates or biological replicates? Was the experiment only performed once? Does this refer to Figure 3?

·      “Although 16 ncRNAs were commonly upregulated in clinical COM tissue and hypoxic KMeC or LMeC versus healthy oral tissue or the relevant normoxic cells, none of them satisfied the criterion for selection as a target ncRNA because they were not upregulated in LMeC (metastatic) vs. KMeC (primary-site) COM cells”. I don't understand why it is important that the ncRNAs had to be upregulated in the metastatic vs primary site cell line?  Especially since only ONE primary vs metastatic cell line was used, it is important not to base all the findings on just this. The other ncRNAs identified merit consideration.

·      I don't understand why Figure 4 and 5 need to be separated? Are the results for Figure 5 not part of Figure 4?

·      What is the logic behind expecting to see ENSCAFT00000084705.1 present in the plasma or in plasma EVs?

Minor points:

·      Line 61: add an “is” (“Moreover, the survival time is reduced to two months…”)

·      Line 78: “….and a full understanding of hypoxic mechanism require evidence on the entire RNA transcriptome” - this part of the sentence needs fixing as it doesn't make sense

·      Figure 3: no statistics have been performed

Author Response

The authors performed NGS on primary and metastatic COM tissues and cell lines to specifically assess hypoxia-mediated non-miRNA ncRNAs, followed by qPCR validation in COM primary and metastatic cells, plasma and EVs. They identified a number of non-miRNA ncRNA species are hypoxically up- or downregulated in COM, and selected the long ncRNA fragment ENSCAFT00000084705.1, as a ‘molecule of interest’ as it was down-regulated in COM tissues, as well as cultured primary and metastatic cell lines (hypoxic and normoxic), when compared to the oral tissues from healthy dogs.

 The manuscript is very well written and the research topic is of interest. However, I have some important concerns.

 Author’s response:

We sincerely appreciate your thorough review of our manuscript and thank you for providing insightful feedback. Your comments have been invaluable in identifying areas that required clarification and improvement. We have carefully addressed each of your suggestions, and we believe these revisions have enhanced the clarity and robustness of our findings.

  • I have concerns with the ‘normal’ tissue samples that were used. They were all from one breed (Beagles) and were from research animals (laboratory animals). Thus, they do not represent appropriate controls for the COM tissue samples which came from dogs of many different breeds and who lived with owners (thus had potential environmental exposures not experienced by the laboratory animals). Appropriate controls would have been normal/healthy tissue from the same animal (such as peri-lesional oral tissue). Were the controls dogs of different ages and sexes as the dogs with COM were? No details have been given.

Response:

We thank the reviewer for identifying an important point. We agree that the tumor-adjacent healthy tissue are the ideal appropriate controls and that tumor-adjacent healthy tissue would be hugely advantageous in evaluating dysregulated ncRNAs from canine oral melanoma patients. Actually, when we initially designed our study, we considered using control tissues (peri-lesional tissue) exactly as the reviewer outlined. However, due to the constraints we ultimately designed our study with beagles from an experimental animal facility as the sources of healthy control tissue.

The tumor tissue and blood samples in this study were collected from dogs with COM undergoing treatment in the Kagoshima University Veterinary Teaching Hospital (KUVTH), and our intuition’s ethics committee judged additional, normal tissue could not be collected from the patients.  Furthermore, distinguishing peritumoral healthy tissue from canine oral melanoma in our biopsy samples and histopathological specimens is challenging for the following reasons. Canine oral melanomas often exhibit infiltrative growth patterns, where tumor cells invade and spread into the surrounding tissues; as such, we could not visually evaluate the tumor-normal tissue's margin. Peritumoral tissues may also appear histologically similar to tumor tissue, especially in cases of inflammation or reactive changes in the surrounding tissue. Based on these constraints, we decided to use beagles as the source of control tissue. We concede that this approach may not have the advantages of the one proposed by the reviewer, but we believe it still allows for comparisons that will reveal target molecules of interest.

  • Only one primary and one metastatic COM cell line were used. There are others available and it is important to show that the findings are not just specific to these two cell lines.

Response:

We agree with the reviewer that more COM cell lines should ideally have been incorporated to validate our NGS findings more robustly. However, we were operating under the constraint of having only one primary (KMeC) and one metastatic (LMeC) COM cell line in our lab, through which we validated our NGS results through qPCR. To overcome this potential limitation, we analyzed the NGS data from at least three biological replicates for each category and confirmed our findings in these COM cell lines. We regard the confirmatory results from the two COM cells as being particularly important at this point. We plan to include additional COM cell lines in our future studies to examine the questions raised by this study, so we believe it is acceptable to use the results from the two cell-line experiments at this stage in our project.

  • Lines 397-90: simply because ENSCAFT00000084705.1 is expressed in humans does not mean there is a possible “link” between canine and human melanomas. This is too much of an extrapolation and this sentence should be removed unless there is any evidence that ENSCAFT00000084705.1 is expressed in human melanoma.

Response:

The authors agree with the reviewer's recommendation to remove the statement suggesting a potential link in the expression of our target lncRNA fragment between canine and human melanoma. While our sequence of interest exhibits similarity to a human lncRNA, this similarity does not imply any strong relationship between human and canine oral melanoma. Furthermore, our study does not explore the role of the targeted lncRNA fragment in human melanoma. Therefore, we removed this statement from the main manuscript. (Line 392-393)

Some clarification is required in places:

  • In addition, the methods state “n = 3 replicates for each cells in either normoxic or hypoxic condition”. Do these represent technical replicates or biological replicates? Was the experiment only performed once? Does this refer to Figure 3?

Response:

Thank you for identifying a poorly phrased point. We included three biological replicates for each cell line under normoxic and hypoxic conditions for NGS analysis. Each replicate was cultured separately and grown independently under the specified conditions. Cell lysates from all replicates were collected as independent samples for subsequent analysis. Furthermore, to ensure reproducibility, cell experiments, and qPCR validations were conducted twice. This additional information has also been incorporated into the main manuscript. (Line 133, 181, 221)

In Figure 3, we depict the expression level of the targeted ncRNA in different NGS samples including COM cells.

  • “Although 16 ncRNAs were commonly upregulated in clinical COM tissue and hypoxic KMeC or LMeC versus healthy oral tissue or the relevant normoxic cells, none of them satisfied the criterion for selection as a target ncRNA because they were not upregulated in LMeC (metastatic) vs. KMeC (primary-site) COM cells”. I don't understand why it is important that the ncRNAs had to be upregulated in the metastatic vs primary site cell line?  Especially since only ONE primary vs metastatic cell line was used, it is important not to base all the findings on just this. The other ncRNAs identified merit consideration.

Response:

In this study, we tried to investigate the common non-miRNA ncRNAs among COM tissue, healthy oral tissue, and hypoxic and normoxic COM cells that are indeed regulated by hypoxic conditions. As described in the introduction, the hypoxic condition is crucial for malignancy; we believe the metastatic condition represents a heightened level of malignancy versus the primary site. Additionally, to enhance the differentiation potential between metastatic and non-metastatic COM, we performed comparisons between LMeC and KMeC. However, we also believe that 16 upregulated ncRNAs which were not common in LMeC vs KMeC, are equally crucial for further investigation, and we plan to target them in future research. However, with limitations such as non-miRNA ncRNA dysregulation in canine tumors, we hope to publish our findings as a novel insight, and address the concerns raised here in future research. This study only focused on examining the expression pattern of one downregulated target.

  • I don't understand why Figure 4 and 5 need to be separated? Are the results for Figure 5 not part of Figure 4?

Response:

Both Figure 4 and Figure 5 present the expression data on ENSCAFT00000084705.1 at various time intervals in hypoxic KMeC and LMeC cells. Figure 4 highlights the expression pattern of the target ncRNA at different time points in each cell type. In contrast, Figure 5 illustrates the target's metastatic potential between the two cell types. Following the reviewer’s suggestion, since both figures are derived from the same experimental data, they have been combined into a single figure, labeled Figure 4. (Line 305)

  • What is the logic behind expecting to see ENSCAFT00000084705.1 present in the plasma or in plasma EVs?

Response:

In this study, we identified a novel hypoxically dysregulated lncRNA fragment that shows differential expression between metastatic and non-metastatic canine oral melanoma, based on NGS and qPCR analysis of tissues and cells. Our interest extended to determining whether this novel fragment is tissue-specific or possesses biomarker potential in clinical situations.

To explore this, we examined the expression of this target in plasma and plasma extracellular vesicles (EVs). However, we found that the expression level of the targeted lncRNA was undetectable in both plasma and plasma EVs from both COM patients and healthy dogs. As a result, we concluded that this lncRNA may be tissue-specific and not likely to be present in extracellular fluids. This result is negative and out of expectation. However, we also would like to enhance this target, which is less promising as a diagnosis blood biomarker. In addition, the discrepancy between tissue and blood containing ncRNA is a well-known phenomenon too.

Minor points:

  • Line 61: add an “is” (“Moreover, the survival time is reduced to two months…”)

Response: We modified in the main manuscript accordingly. (Line 52)

  • Line 78: “….and a full understanding of hypoxic mechanism require evidence on the entire RNA transcriptome” - this part of the sentence needs fixing as it doesn't make sense

Response: We modified in the main manuscript accordingly. (Line 77-79)

  • Figure 3: no statistics have been performed

Response:

Figure 3 depicts the expression levels of the ncRNA target across various NGS samples. Mostly, we utilized three (n=3) samples in each NGS category, which needs to be improved for statistical analysis. Therefore, statistical analysis was not performed for Figure 3.

Reviewer 2 Report

Comments and Suggestions for Authors

In this study, Hino et al aimed to elucidate non-miRNA ncRNAs that may be mediated by hypoxia in canine oral melanoma tumors, primary and metastatic cell lines as compared to heathy tissues. The authors identified one downregulated ncRNA with potential as a therapeutic or diagnostic biomarker; however, it was not detectable in plasma and plasma EV samples.

Major comments

1) a) Section 2.1: Please include a Table with patient characteristics, including dog breed, age, sex, previous or current treatments, tumor histopathological charasteristics. 

b) Please add a relevant paragraph at the Results section before section 3.1.

c) NcRNA expression has high tissue specificity. Please specify the area that was sampled from the healthy animals and was used as control tissue for comparison with melanoma tumors and melanoma cell lines. Melanoma cell lines are derived from the isolation of neoplastic melanocytes, so the appropriate control would be to use isolated normal melanocytes.  

d) Were the control dogs matched with the ones with COM with respect to age and breed?

e) The authors indicate that the control group consisted of dogs that were bred for research. These dogs are housed under strict conditions, unlike pet dogs and factors such as the gut microbiome and subsequent physiological and pathological processes of the host dogs may be vastly different. All the abovementioned factors could potentially affect the expression of ncRNAs.

2) Section 2.8: The authors used non- parametric statistical tests to assess statistical significance. Did the authors check whether the assumptions of the equivalent parametric tests were met (such as normal distribution or equality of variance etc) before using a non- parametric test?

3) Line 271: The authors state that the 16 upregulated ncRNAs identified were not studies further because they were not upregulated in metastatic vs primary site-derived melanoma cell lines. Instead, they decided to focus on 1 downregulated transcript based on similar comparisons. However, the aim of the paper is to focus on hypoxia-mediated transcriptome as compared to normoxia, regardless of the localization of the tumor. As a result, all these ncRNAs should have been addressed further. Please address this discrepancy. Also, from a clinical standpoint, it is easier to detect upregulated than downregulated biomarkers (that may be undetectable like the selected ncRNA of this article) so these 16 transcripts could be of higher clinical interest. 

Minor comments

4) The 2nd paragraph of the Introduction section (lines 57-66) could be used as the 1rst paragraph in this section.

5) Line 48: "major killer of dogs": Please rephrase

6) Line 75: Please remove "and" before lncRNAs

7) line 88: "was then to have its expression": Please rephrase

8) Line 107: Please briefly describe the specific tubes and the method used to obtain blood plasma.

9) Line 116-117: Please add the percentages/concentrations of L-glutamine and antibiotics added in the culture medium.

10) Line 204: Did you use "First" or "fast" advanced Master Mix kit? Please correct if needed and add the catalog number.

11) Section 3.1 and Figure 1: Please specify whether the incubation time for the hypoxic cell lines that were submitted for NGS profiling

12) Figure 2: The numbers in the Venn Diagrams are very small. Please increase the font.

Comments on the Quality of English Language

There are several grammatical errors that need to be addressed. 

Author Response

In this study, Hino et al aimed to elucidate non-miRNA ncRNAs that may be mediated by hypoxia in canine oral melanoma tumors, primary and metastatic cell lines as compared to heathy tissues. The authors identified one downregulated ncRNA with potential as a therapeutic or diagnostic biomarker; however, it was not detectable in plasma and plasma EV samples.

Author’s response:

We sincerely appreciate your thorough review of our manuscript and for providing insightful feedback. Your comments have been invaluable in identifying areas that required clarification and improvement. We have carefully addressed each of your suggestions, and these revisions have enhanced the clarity and robustness of our findings.

Major comments

  1. a) Section 2.1: Please include a Table with patient characteristics, including dog breed, age, sex, previous or current treatments, tumor histopathological charasteristics. 

Response:

We thank the reviewer for pointing out information that is needed, and we agree that this information is necessary for proper clarification of the clinical samples. Therefore, we have added all clinical sample information in tabular form in the main manuscript as Table 1. (Line 106-107)

Table 1. Canine oral melanoma tissue and plasma sample information.

No

Age (Years)

Sex

Breed

Tumor Stage

Metastasis Status

Types of Sample

1

13.3

Male

M.D.

I

Tissue, Plasma

2

10.3

Male

Yorkshire

IV

Tissue, Plasma

3

10.2

Male

Chiwawa

IV

P

Tissue, Plasma

4

12.7

Male

M.D.

IV

P

Tissue, Plasma

5

14.8

Male

Mongrel

IV

P

Tissue, Plasma

6

10

Male

Golden Retriever

IV

Tissue, Plasma

7

10.11

Male

M.D.

I

Tissue

8

7.11

Male

M.D.

I

Tissue, Plasma

9

10.9

Male

M.D.

IV

Tissue, Plasma

10

12

Male

Shiba

IV

P

Tissue

11

13

Male

Pomerania

I

Tissue

12

16.3

Male

M.D.

IV

P

Tissue, Plasma

13

11

Male

M.D.

IV

P

Tissue, Plasma

14

12

Male

Mongrel

I

Tissue, Plasma

15

11.1

Male

M.D.

IV

P

Tissue, Plasma

16

15.6

Male

Pomeranian

II

P

Tissue, Plasma

17

12.11

Male

M.D.

IV

P

Tissue

18

12.4

Male

Shiba

IV

P

Tissue

19

10.8

Male

M.D.

IV

Tissue

20

15.2

Male

Shiba

I

Tissue

21

12.4

Female

M.D.

IV

Tissue, Plasma

22

14.6

Female

M.D.

II

P

Tissue

23

15.2

Female

Mongrel

IV

Tissue

24

15.2

Female

Mongrel

IV

P

Tissue

25

8.2

Female

M.D.

IV

P

Tissue, Plasma

26

15.3

Female

Mongrel

I

Tissue, Plasma

27

15.3

Female

Mongrel

I

Tissue, Plasma

28

11.8

Female

M.D.

I

Tissue, Plasma

29

14

Female

Dalmatian

II

P

Tissue, Plasma

30

12.1

Female

Toy poodle

IV

P

Tissue, Plasma

(M.D.) indicates “Miniature Dachshund”, (P) indicates “Present”, and (−) indicates “Absent.”

  1. b) Please add a relevant paragraph at the Results section before section 3.1.

Response:

We would like to check our understanding of “relevant paragraph”; would this refer to a paragraph describing the information in Table 1? We have considered that, but we believe that as the clinical samples were selected at the start of the study, it could be argue that the study population is not a result, and that we can introduce it with the text we have in the Methods and Materials section.

  1. c) NcRNA expression has high tissue specificity. Please specify the area that was sampled from the healthy animals and was used as control tissue for comparison with melanoma tumors and melanoma cell lines. Melanoma cell lines are derived from the isolation of neoplastic melanocytes, so the appropriate control would be to use isolated normal melanocytes.  

Response:

For NGS, we obtained three control samples of healthy oral mucosal tissue from the beagle breed for experimental purposes at an animal research facility, following thorough examination and investigation. Control tissues were sampled from locations corresponding to where lesions most frequently occur. The COM cells used in this study were provided by Dr. Takayuki Nakagawa of Tokyo University and cultured according to established methods and protocols in the published literature on these cell lines. As the reviewer suggested, the control melanocyte cell line is the most appropriate control, but no established dog melanocyte cell line exists. So, we opted not to use the melanocyte cells as a control. We then also confirmed that the expression reflected the hypoxic condition in a time-dependent manner in cultural cells.

  1. d) Were the control dogs matched with the ones with COM with respect to age and breed?

Response:

Unlike the COM patients, all healthy controls were selected from a single breed and within a certain age range. We agree this as a limitation of our study and have added the following sentences in the limitation section of our main manuscript “Furthermore, our control dogs were not matched by age, sex, or breed with the main study population. Further studies are needed to validate the clinical usefulness with an increased sample size in which control dogs reflect the age range, sex distribution, and range of breeds in the tumor-bearing population.” (Line 420-423)

  1. e) The authors indicate that the control group consisted of dogs that were bred for research. These dogs are housed under strict conditions, unlike pet dogs and factors such as the gut microbiome and subsequent physiological and pathological processes of the host dogs may be vastly different. All the abovementioned factors could potentially affect the expression of ncRNAs.

Response:

We agree with this statement. We agree that the tumor-adjacent healthy tissue are the ideal appropriate controls and that tumor-adjacent healthy tissue would be hugely advantageous in evaluating dysregulated ncRNAs from canine oral melanoma patients. Actually, when we initially designed our study, we considered using control tissues (peri-lesional tissue) exactly as the reviewer outlined. However, due to the constraints we ultimately designed our study with beagles from an experimental animal facility as the sources of healthy control tissue.

The tumor tissue and blood samples in this study were collected from dogs with COM undergoing treatment in the Kagoshima University Veterinary Teaching Hospital (KUVTH), and our intuition’s ethics committee judged additional, normal tissue could not be collected from the patients.  Furthermore, distinguishing peritumoral healthy tissue from canine oral melanoma in our biopsy samples and histopathological specimens is challenging for the following reasons. Canine oral melanomas often exhibit infiltrative growth patterns, where tumor cells invade and spread into the surrounding tissues; as such, we could not visually evaluate the tumor-normal tissue's margin. Peritumoral tissues may also appear histologically similar to tumor tissue, especially in cases of inflammation or reactive changes in the surrounding tissue. Based on these constraints, we decided to use beagles as the source of control tissue. We concede that this approach may not have the advantages of the one proposed by the reviewer, but we believe it still allows for comparisons that will reveal target molecules of interest.

We also believe that natural conditions may affect the ncRNA expression in tissues. Therefore, we focused on the hypoxic conditions and represented their stimulation in a cell culture study. We think that we can at least say that the target ncRNA in this study is affected by hypoxic conditions, one of the major factors of the tumor microenvironment and malignancy.

2) Section 2.8: The authors used non- parametric statistical tests to assess statistical significance. Did the authors check whether the assumptions of the equivalent parametric tests were met (such as normal distribution or equality of variance etc) before using a non- parametric test?

Response:

We checked whether the data is normally distributed or not using D’Agostino-Pearson omnibus normality test and Shapiro-Wilk normality test and found all datasets are normally distributed (p > 0.05). However, the authors decided to use non-parametric statistical tests due to the comparatively small sample size [only six (n=6) biological replicates of cells in each sample category]. Parametric tests often require a larger sample size to achieve adequate statistical power. We followed recommendations that at least 30 samples be collected per group to ensure the validity of the parametric test results. With only six observations per group, the normality assumption may not hold robustly. That is why we used a non-parametric test for statistical analysis. 

3) Line 271: The authors state that the 16 upregulated ncRNAs identified were not studies further because they were not upregulated in metastatic vs primary site-derived melanoma cell lines. Instead, they decided to focus on 1 downregulated transcript based on similar comparisons. However, the aim of the paper is to focus on hypoxia-mediated transcriptome as compared to normoxia, regardless of the localization of the tumor. As a result, all these ncRNAs should have been addressed further. Please address this discrepancy. Also, from a clinical standpoint, it is easier to detect upregulated than downregulated biomarkers (that may be undetectable like the selected ncRNA of this article) so these 16 transcripts could be of higher clinical interest. 

Response:

In this study, we tried to investigate the common non-miRNA ncRNAs among COM tissue, healthy oral tissue, and hypoxic and normoxic COM cells that are regulated by hypoxic conditions. As described in the introduction, the hypoxic condition plays crucial role in progression to malignancy; we believe the metastatic condition represents a higher level of malignancy than the primary site. Additionally, to enhance the differentiation potential between metastatic and non-metastatic COM, we performed comparisons between LMeC and KMeC. However, we also believe that 16 upregulated ncRNAs, which were not common in LMeC vs KMeC, are equally valuable targets for further investigation, and we plan to investigate them. However, with limitations such as non-miRNA ncRNA dysregulation in canine tumors, we hope to publish our findings as a novel insight, and address the concerns raised here in future research.  This study only focused on examining the expression pattern of one downregulated target.

Minor comments

4) The 2nd paragraph of the Introduction section (lines 57-66) could be used as the 1rst paragraph in this section.

Response:

We have replaced the paragraph as you recommend. (Line 48-66)

5) Line 48: "major killer of dogs": Please rephrase

Response: This statement is rephrased in the manuscript. (Line 58)

6) Line 75: Please remove "and" before lncRNAs

Response: “and” is removed from the manuscript. (Line 74-75)

7) line 88: "was then to have its expression": Please rephrase

Response: This statement is rephrased in the manuscript. (Line 87-88)

8) Line 107: Please briefly describe the specific tubes and the method used to obtain blood plasma.

Response:

The method for collection of blood samples and preparation of plasma from blood samples is described in the manuscript. (Line 111-116)

“Blood samples were collected in tubes containing 3.2% sodium citrate anticoagulant and then centrifuged for 10 minutes at 3000 × g immediately after collection to separate plasma from other cellular elements such as red blood cells (RBCs). The supernatant was then carefully aspirated and subjected to additional high-speed centrifugation at 16,000 × g for 10 minutes at 4°C to remove residual cellular debris or platelets. Pure plasma was collected from the supernatant without disturbing the pellet.”

9) Line 116-117: Please add the percentages/concentrations of L-glutamine and antibiotics added in the culture medium.

Response: We have added these at the appropriate point. (Line 123-125)

10) Line 204: Did you use "First" or "fast" advanced Master Mix kit? Please correct if needed and add the catalog number.

Response:

The authors apologize for the error. The correct name is "TaqMan Fast Advanced Master Mix." This correction has been made in the main manuscript. Additionally, the catalog number has been included. (Line 213-214)

11) Section 3.1 and Figure 1: Please specify whether the incubation time for the hypoxic cell lines that were submitted for NGS profiling

Response:

We submitted hypoxic cells that were incubated for 48 hours for NGS. This information is added in the main manuscript. (Line 182)

12) Figure 2: The numbers in the Venn Diagrams are very small. Please increase the font.

Response:

We have increased the font size of the numbers in Figure 2. (Line 271)

Reviewer 3 Report

Comments and Suggestions for Authors

In general, the manuscript is written clearly and describes the methodology in detail. The review link has some marked questions that I responded to. Below, I answer the questions placed by the editor:   

The manuscript is a study that refers to the isolation of a long non-coding RNA fragment named ENSCAFT00000084705.1 which is a molecule related to hypoxia present in canine oral melanoma and its metastases.

The relevance of the article is related to the fact that the isolation of RNA fragments indicates the role of hypoxia in canine oral melanoma (COM). Hypoxia represents a way of development of neoplasm, contributing to the dissemination of malignant cells. The manuscript refers to the relevance of this and analyses the role of the ENSCAFT00000084705.1  in the COM.

The study identifies the ncRNA molecule (ENSCAFT00000084705.1) indicating its relation with hypoxia in COM

In my opinion the methodology is well describe, I do not add any question

The conclusion is consistent in relation with the arguments present. The study is very specific in the idea of identifying the ncRNA molecule and expressing its relationship with COM and metastases doing a comparison with normal oral tissue.

References are appropriate, I only recommended reinforcing the discussion with more references and include in paragraphs that require references for example in lines 341 to 344 and as previously indicated avoid using a first or third person in writing and avoid repetitive self-citation.   

I don’t have any comments for figures or data.  The supplementary table is very difficult to interpret, but it is only for supplementary data, so I think that it is not necessary to make any comments about it.

Line 48: please restructure this sentence “Canine oral melanoma is a major killer of dogs, and”……, a suggestion is “Canine oral melanoma is one of the most aggressive tumors in dogs”………..

Author Response

In general, the manuscript is written clearly and describes the methodology in detail. The review link has some marked questions that I responded to. Below, I answer the questions placed by the editor:   

The manuscript is a study that refers to the isolation of a long non-coding RNA fragment named ENSCAFT00000084705.1 which is a molecule related to hypoxia present in canine oral melanoma and its metastases.

The relevance of the article is related to the fact that the isolation of RNA fragments indicates the role of hypoxia in canine oral melanoma (COM). Hypoxia represents a way of development of neoplasm, contributing to the dissemination of malignant cells. The manuscript refers to the relevance of this and analyses the role of the ENSCAFT00000084705.1 in the COM.

The study identifies the ncRNA molecule (ENSCAFT00000084705.1) indicating its relation with hypoxia in COM.

In my opinion the methodology is well describe, I do not add any question

The conclusion is consistent in relation with the arguments present. The study is very specific in the idea of identifying the ncRNA molecule and expressing its relationship with COM and metastases doing a comparison with normal oral tissue.

Author’s response:

We sincerely express our gratitude for your appreciation and kind words regarding this manuscript. We also appreciate your thoughtful review of our manuscript and for providing insightful feedback. We have carefully addressed each of your suggestions, and we believe these revisions have enhanced the clarity and robustness of our findings.

References are appropriate, I only recommended reinforcing the discussion with more references and include in paragraphs that require references for example in lines 341 to 344 and as previously indicated avoid using a first or third person in writing and avoid repetitive self-citation. 

Response:

We have revised the discussion section in accordance with the reviewer’s suggestion in the main manuscript.

I don’t have any comments for figures or data.  The supplementary table is very difficult to interpret, but it is only for supplementary data, so I think that it is not necessary to make any comments about it.

Line 48: please restructure this sentence “Canine oral melanoma is a major killer of dogs, and”……, a suggestion is “Canine oral melanoma is one of the most aggressive tumors in dogs”………..

 Response:

The statement is rephrased in the main manuscript according to the suggestion. (Line: 58-59)

Round 2

Reviewer 1 Report

Comments and Suggestions for Authors

Thank you for your answers to my queries and for considering my suggestions.

Reviewer 2 Report

Comments and Suggestions for Authors

I would like to thank the authors for taking my comments into consideration and significantly improving the manuscript. The methods and results are adequately described, and the limitations of the study are clearly defined.